# A high capacity small molecule quinone cathode for rechargeable aqueous zinc-organic batteries

Zirui Lin [1], Hua-Yu Shi[1], Lu Lin[1], Xianpeng Yang[1], Wanlong Wu[1] & Xiaoqi Sun [1✉]

Rechargeable aqueous zinc-organic batteries are promising energy storage systems with low-cost aqueous electrolyte and zinc metal anode. The electrochemical properties can be systematically adjusted with molecular design on organic cathode materials. Herein, we use a symmetric small molecule quinone cathode, tetraamino-p-benzoquinone (TABQ), with desirable functional groups to protonate and accomplish dominated proton insertion from weakly acidic zinc electrolyte. The hydrogen bonding network formed with carbonyl and amino groups on the TABQ molecules allows facile proton conduction through the Grotthuss-type mechanism. It guarantees activation energies below 300 meV for charge transfer and proton diffusion. The TABQ cathode delivers a high capacity of 303 mAh g$^{-1}$ at 0.1 A g$^{-1}$ in a zinc-organic battery. With the increase of current density to 5 A g$^{-1}$, 213 mAh g$^{-1}$ capacity is still preserved with stable cycling for 1000 times. Our work proposes an effective approach towards high performance organic electrode materials.

[1] Department of Chemistry, Northeastern University, Shenyang, China. ✉email: sunxiaoqi@mail.neu.edu.cn

Rechargeable lithium-ion batteries have been widely used in portable electronics and electric vehicles[1–5]. However, cost and potential safety issues restrict their application areas and scales[6–10]. Rechargeable aqueous batteries using aqueous electrolytes largely reduce those concerns and experience rapid development in the past few years. The zinc metal is a desirable anode for aqueous systems and delivers high theoretical specific capacity (820 mAh g$^{-1}$), low redox potential (-0.76 V vs. S.H.E.) as well as good compatibility with water[11–13]. The research on suitable cathode materials to couple with the zinc anode mainly focuses on inorganic compounds such as metal oxides, Prussian blue salts and polyanion compounds[14–17]. Among them, manganese oxides and vanadium oxides deliver high capacities;[18–21] however, problems such as active material dissolution are important challenges for their further applications[14].

In comparison to inorganic materials, organic compounds are less explored as cathode materials for aqueous batteries[22]. Nevertheless, organic compounds are more flexible with molecular design, which allows the systematic adjustment of voltage, capacity, conductivity, redox kinetics and other properties for electrode materials[23–25]. For example, Koshika et al. attached the electrochemically active 2,2,6,6-tetramethylpiperidine-1-oxyl to the poly(vinyl ether) chain in order to avoid dissolution. The obtained electrode delivers 131 mAh g$^{-1}$ capacity and excellent rate capability with the facile radical-involved redox process[26]. Our group showed that sulfo group can be introduced to the six-membered carbon ring of polyaniline to function as an internal proton reservoir and retain a locally high acidic environment. The electrochemical activity of polyaniline is therefore maintained in the weakly acidic Zn electrolyte[27].

Among organic materials, quinone compounds show good environmental friendliness. They widely exist in nature, and the redox process between quinone and hydroquinone is an important physiological progress in living organisms[28]. The excellent redox activity of quinones also makes them desirable cathode materials for aqueous zinc batteries. Zhao et al. reported a calix[4] quinone (C4Q) molecule, where each benzoquinone unit is connected by C-C single bond to form a bowl-like structure. The cathode displays a high capacity of 335 mAh g$^{-1}$ at 20 mA g$^{-1}$ [28]. Nam and co-workers designed a medium molecular weight quinone compound, triangular macrocyclic phenanthrenequinone (PQ-Δ), which allows the co-insertion of Zn$^{2+}$ and water and delivers a specific capacity of 225 mAh g$^{-1}$ at 30 mA g$^{-1}$ [29]. Guo et al. proposed a pyrene-4,5,9,10-tetraone (PTO) cathode, achieving 336 mAh g$^{-1}$ capacity at 40 mA g$^{-1}$ [30].

Zn$^{2+}$ de/intercalation is demonstrated for most quinone materials in zinc cells; however, the diffusion of multivalent cation could be sluggish[31]. On the other hand, the facile insertion of proton into cathode materials from weakly acidic zinc electrolytes is possible with the hydrolysis of Zn$^{2+}$ to continuously generate proton[31–33], and proton could take part in the redox process of quinones[34–36]. Besides, most of the previously studied quinone-based compounds are relatively complex in molecular structures and require multi-step synthesis procedures. In comparison, small molecule quinones provide the advantage of simple synthesis, corresponding to a reduced cost. However, many small quinone molecules, such as benzoquinone, are prone to sublimation. It leads to difficulties in electrode manufacturing, handling and storage. Another important concern of small molecules is their relatively high solubility, resulting in the shuttling of active material in electrolytes and capacity decay. Nevertheless, previous studies have shown that molecules with high symmetry possess low dipole moments and thus low solubilities in aqueous solutions[37]. Considering the above factors, we herein specifically apply a symmetric small molecule of tetraamino-p-benzoquinone (TABQ) as the cathode material for aqueous zinc-organic batteries. The four amino groups are available for protonation in weakly acidic zinc electrolytes, which would create a proton active environment and initiate proton insertion. At the same time, hydrogen bonds are formed among amino and carbonyl groups, so that the compound does not sublimate even at elevated temperatures. More importantly, the hydrogen bonding network allows facile proton conduction through the breaking and reforming of hydrogen bonds. The TABQ electrode thus functions with dominated proton de/intercalation and experiences low activation energies for charge transfer and diffusion. Excellent electrochemical performance is achieved.

## Results

The TABQ active material is soluble in N-Methyl-2-pyrrolidone (NMP), and it undergoes a dissolution-reprecipitation process during electrode manufacturing. Figure 1a shows the Fourier transform infrared (FT-IR) spectrum of the TABQ active material after drying out from NMP (no carbon and binder). The multiple absorption peaks between 3469 and 3262 cm$^{-1}$ (purple part) belong to the N-H stretching vibrations with different symmetry modes from the four amino groups in TABQ. Notably, a broad band shows up on the right side as highlighted in cyan. It suggests the redshift of part of the N-H stretching vibration peak, which is a typical behavior upon the formation of hydrogen bonds[38,39]. Another site available for forming hydrogen bonds with amino is oxygen on carbonyl in the case of TABQ. Correspondingly, the carbonyl stretching vibration at 1547 cm$^{-1}$ in TABQ (pink part) experiences more than 100 cm$^{-1}$ of redshift in comparison to 1661 cm$^{-1}$ in benzoquinone (spectral database for organic compounds SDBS). Although the electron-donor property of amino also causes the redshift of carbonyl, such a significant shift results from additional effects, i.e., the hydrogen bonding[37,40]. Finally, the peaks in the range of 1700–1270 cm$^{-1}$ shown in green belong to the vibrations of the six-membered carbon ring containing carbon–carbon single bonds, double bonds and their synergistic effects.

The above analysis on FT-IR demonstrates the formation of hydrogen bonds with the amino and carbonyl functional groups in TABQ molecules. It results in strong interactions between adjacent molecules and furthermore affects the physical and chemical properties. We studied the sublimation behavior of TABQ using the setup shown in Supplementary Fig. 1, and benzoquinone was applied as the comparing standard. When the benzoquinone powder was heated at 90 °C for 3 h in the oven, no original powder is left and yellow crystals are condensed all over the flask which stayed at room temperature (Fig. 1b). It demonstrates the sublimation of benzoquinone. With TABQ, on the contrary, the original powder remains on the aluminum cover which stayed in the oven and the flask is completely clear (Fig. 1c). The same phenomenon is observed even with more elevated temperature of 150 °C (Fig. 1d). In addition, thermogravimetric analysis (TGA) with constant temperature hold at 90 °C shows continuous weight loss of benzoquinone as a result of sublimation, whereas a stable weight evolution is obtained with TABQ (Fig. 1e). The above analysis confirms the introduction of hydrogen bonds effectively suppresses the sublimation of small molecule quinone. It would provide convenience for various electrode treatments.

The solubility of TABQ in 1 M ZnSO$_4$, which is a conventional electrolyte for zinc batteries, was quantified by UV–vis analysis (Fig. 1f)[37]. The saturated concentration was determined to be 1.7 mmol L$^{-1}$. This low solubility would ensure stable cycling in aqueous zinc cells. The relationship between molecular structure and solubility is further studied by comparing with related

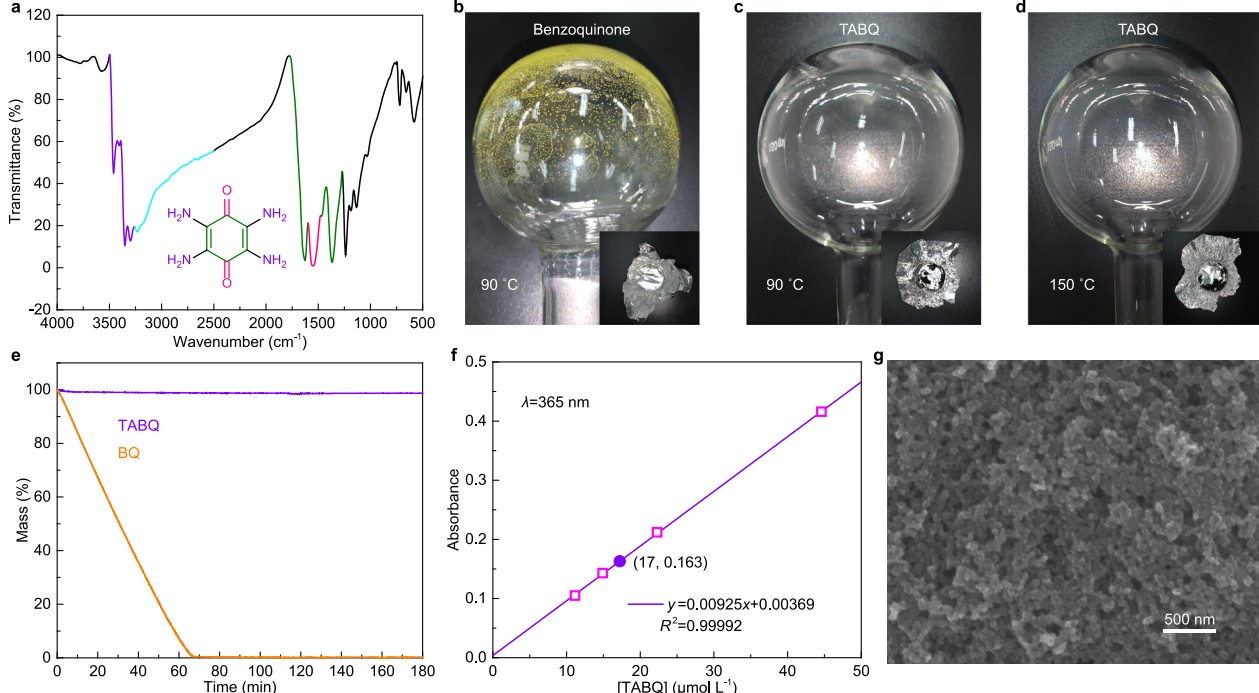

**Fig. 1 Characterizations of TABQ. a** FT-IR spectrum of TABQ. **b**, **c**, **d** Sublimation test of benzoquinone and TABQ at 90 °C and 150 °C. **e** TGA curves of benzoquinone and TABQ with constant temperature hold at 90 °C. **f** UV-vis calibration curve of TABQ in 1 M ZnSO$_4$ together with the calculation of the 100 times diluted saturated solution. **g** SEM image of the TABQ electrode.

compounds of tetrahydroxy-p-benzoquinone (THBQ), tetrachloro-o-benzoquinone (o-TCBQ) and tetrachloro-p-benzoquinone (p-TCBQ). UV–vis analysis results in the solubilities of 3.4 mmol L$^{-1}$ and 2.3 mmol L$^{-1}$ for THBQ and o-TCBQ, respectively (Supplementary Fig. 2), and very trace dissolution of p-TCBQ is noticed. Literature also demonstrated the insoluble nature of p-TCBQ[41]. Comparing among the p-quinones, the higher solubilities of TABQ and THBQ than p-TCBQ should be attributed to the more hydrophilic amino or hydroxyl functional groups. Nevertheless, the solubilities are still quite low thanks to their highly symmetric molecular structures and thus low dipole moments[37]. For example, the p-TCBQ molecule is more symmetric than o-TCBQ and presents lower solubility.

Figure 1g shows the scanning electron microscopy (SEM) image of the TABQ electrode. The TABQ active material forms as thin layer and is homogeneously distributed over Ketjen Black (KB) spherical particles, resulting in a conductive network with porous structure. The effective interaction between TABQ and carbon would ensure good electrical conductivity in electrodes. Comparing with the high crystallinity of as-prepared TABQ, the X-ray diffraction (XRD) pattern of TABQ electrode shows broad peaks due to short coherent lengths in the thin layer (Supplementary Fig. 3). The diffractions can be attributed to the parallel stacking of six-membered carbon rings in the quinone structure[41].

The electrochemical performance of the TABQ cathode was tested in zinc-organic cells with zinc foil anode and 1 M ZnSO$_4$ aqueous electrolyte. The two most commonly used salts in zinc electrolytes are ZnSO$_4$ and Zn(CF$_3$SO$_3$)$_2$. They provide similar performance for TABQ (discussed later), and ZnSO$_4$ was selected considering the lower price. Figure 2a, b shows the voltage profiles and capacity evolution at various current densities. The cathode delivers 303 mAh g$^{-1}$ capacity at the current density of 0.1 A g$^{-1}$. The Ketjen Black (KB) conductive agent provides less than 25 mAh g$^{-1}$ capacity without any redox peaks in the differential capacity curve (Supplementary Fig. 4), demonstrating the capacity

is majorly contributed by the TABQ active material. With the increase of current density to 5 A g$^{-1}$, the TABQ cathode still maintains 213 mAh g$^{-1}$ capacity. It suggests the excellent electrochemical activity of both carbonyl groups on TABQ. Figure 2c shows the differential capacity curve at 0.1 A g$^{-1}$. Two pairs of redox peaks are observed at 0.90 V/1.02 V and 0.77 V/0.83 V, respectively. The small overpotential demonstrates the good cation de/insertion kinetics into TABQ. The electrochemical performance of TABQ is compared with previously reported quinone cathode materials in zinc-organic batteries, including C4Q[28], PQ-Δ[29], PTO[30], TCBQ[41], HqTp[42], poly(benzoquinonyl sulfide) (PBQS)[43] and dibenzo[b,i]thianthrene-5,7,12,14-tetraone (DTT)[44]. As shown in Fig. 2d, TABQ delivers the highest capacity at 0.1 A g$^{-1}$ and presents the best rate capability. Figure 2e and Supplementary Fig. 5 show the long-term cycling behavior. At the current density of 5 A g$^{-1}$, the TABQ cathode exhibits stable capacity retention for over 1000 cycles after slight capacity decay during the first few cycles. The coulombic efficiencies are close to 100%. FT-IR and SEM analysis on the electrode after 1000 cycles verifies the excellent compositional and morphological stabilities of TABQ (Supplementary Fig. 6), which ensures the superior cycling performance.

The stability of TABQ in zinc cells was further confirmed by a rest test, where the cell was cycled and rested for 10 h after discharged to half capacity, fully discharged, charged to half capacity and fully charged, respectively (Supplementary Fig. 7a). It corresponds to a total of 40 h rest period, allowing any possible dissolution to take place. Supplementary Fig. 7b–d shows the capacity retention, charge-discharge and differential capacity curves during the test. They are well preserved before, during and after the rest cycle, verifying the excellent stability of TABQ at various states. Another Zn-TABQ cell was assembled with the electrolyte of 1 M ZnSO$_4$ containing saturated TABQ and cycled at 0.1 A g$^{-1}$. The anode does not show any nitrogen signal by X-ray photoelectron spectroscopy (XPS) after 10 cycles (Supplementary Fig. 8). It demonstrates that the small amount of TABQ dissolved in electrolyte does not react with Zn and there is no

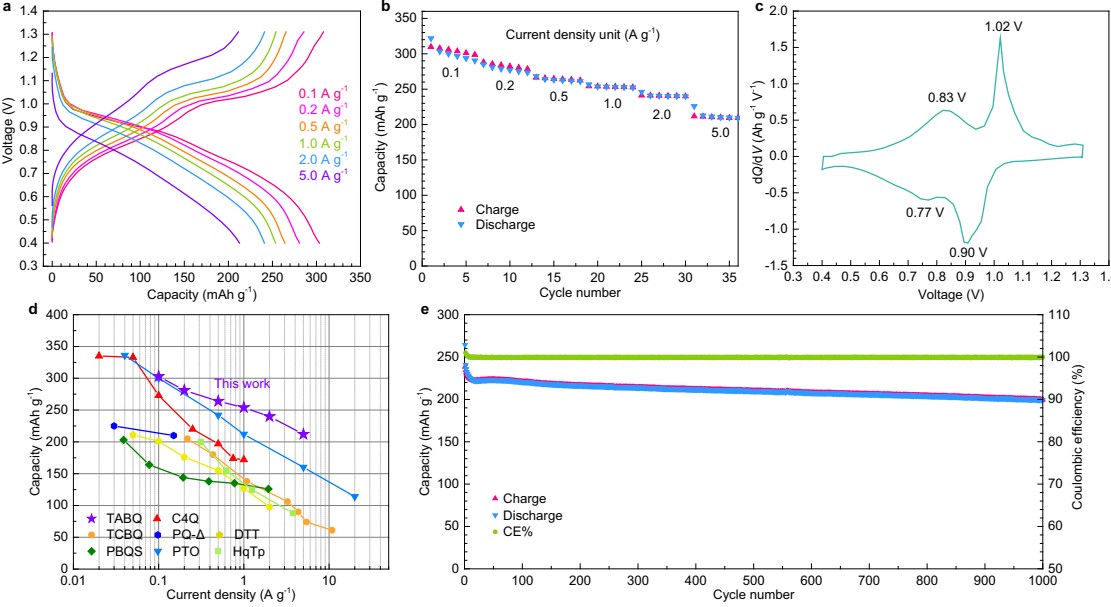

**Fig. 2 Electrochemical performance of the TABQ cathode in aqueous zinc-organic batteries. a** Charge/discharge curves and **b** capacity evolution under various current densities. **c** Differential capacity curve at 0.1 A g⁻¹. **d** Comparison with previously reported quinone materials. **e** Capacity and coulombic efficiency evolution at 5 A g⁻¹ for 1000 cycles.

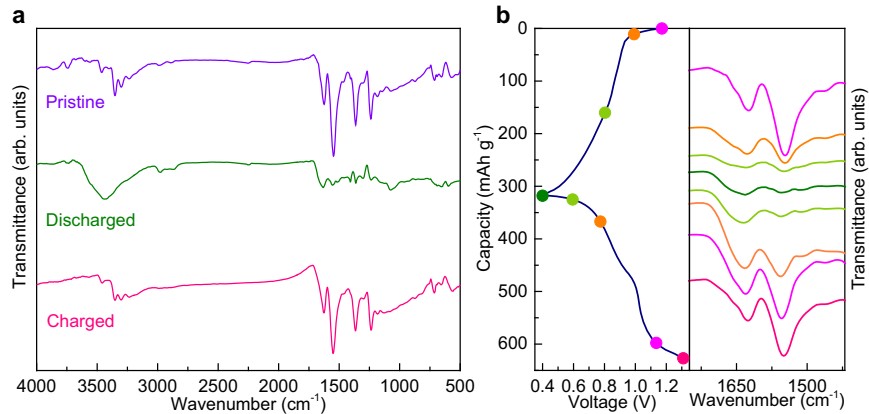

**Fig. 3 Ex-situ FT-IR characterizations of TABQ. a** FT-IR spectra of the TABQ electrode at the pristine, discharged and charged states. **b** Detailed FT-IR peak evolution along discharge and charge.

shuttling of TABQ. Overall, the excellent cycling stability of Zn-TABQ cells is confirmed.

The evolution of TABQ during charge and discharge was studied by ex-situ FT-IR (Fig. 3a). Upon discharge, the carbonyl peak at 1547 cm⁻¹ disappears, and the stretching vibration of carbon-carbon double bond at 1627 cm⁻¹ shows much reduced intensity. The disappearance of carbonyl vibration is a result of its reduction during the discharge process. More specifically, -C=O accepts electron and transforms into -C-O⁻ anion. Simultaneously, cations insert to compensate the negative charge[34–36,45]. This reduction process also converts the conjugated quinone ring to the π-conjugated benzene ring. Since the quinone structure exhibits symmetric double bonds in the six-membered carbon ring while the benzene ring does not, the former provides larger dipole moment change of anti-symmetric stretching vibration and therefore shows stronger IR absorption. Upon charge, the FT-IR spectrum resembles the one for the pristine electrode, confirming the reversible redox of carbonyl group and transition between conjugated structures. Figure 3b shows the more detailed peak evolution with measurements taken at various states along discharge and charge.

In addition, a periodic peak shape change is observed in the range of 3000–4000 cm⁻¹ (Fig. 3a). At the end of discharge, a broad band appears and overlaps the original peaks of amino groups. The band disappears and the amino vibrations are revealed upon charge. The broad band is the characteristic absorption of water. The electrode was further studied to track its origin. Figure 4a shows the XRD of the electrode at different states. The change on the TABQ diffractions results from the transformation between conjugated quinone rings and π-conjugated benzene rings which exhibit different stacking distances[41]. The narrower peaks in the charged electrode suggest an enhanced long-range ordering. More significantly, the diffraction peak of $Zn_4SO_4(OH)_6 \cdot 4H_2O$ shows up in the discharged electrode, and it disappears upon charge. The same change is also noted in an in-situ XRD cell (Supplementary Fig. 9), suggesting the reversible formation of $Zn_4SO_4(OH)_6 \cdot 4H_2O$ at the cathode during the electrochemical processes. The SEM image of the electrode shows the reversible appearance and disappearance of platelets over the pristine nano-particles upon discharge and charge (Fig. 4b), which is the typical morphology of zinc sulfate

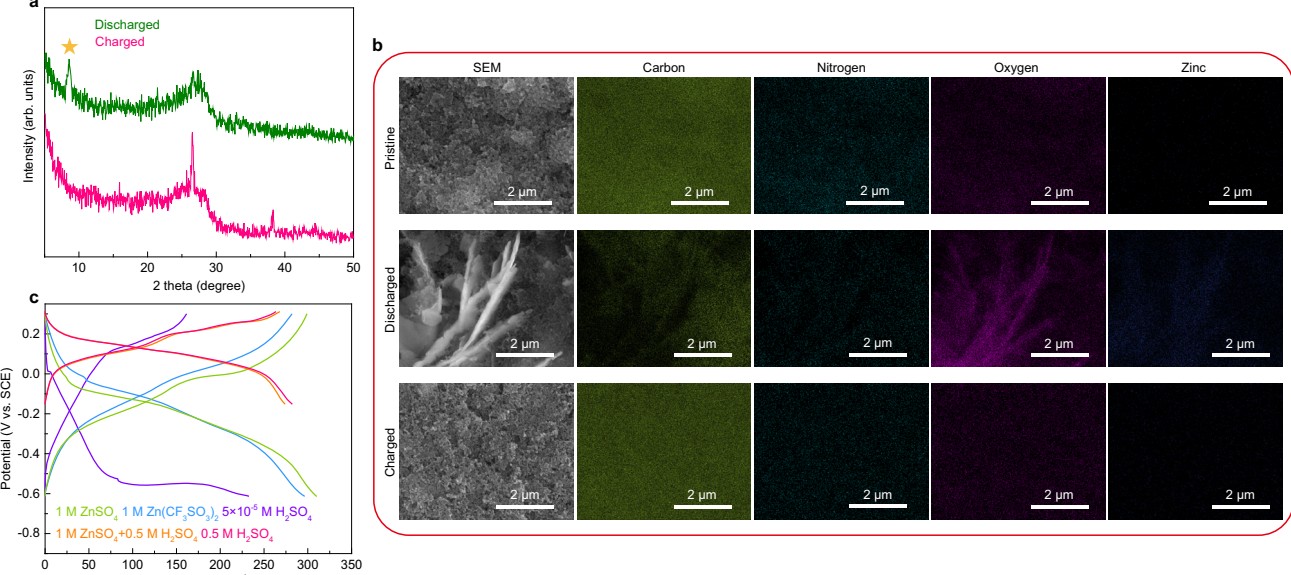

**Fig. 4 Proof of proton insertion in TABQ. a** XRD of TABQ at the end of discharge and charge (the asterisk shows the characteristic diffraction peak of $Zn_4SO_4(OH)_6\cdot 4H_2O$). **b** SEM images and EDS mappings of TABQ at different states. **c** Charge/discharge curves of TABQ in three electrode cells with SCE reference electrode and different electrolytes.

hydroxide[19]. Energy dispersive X-ray spectroscopy (EDS) analysis gives a Zn:S ratio around 4 in the discharged electrode (Supplementary Fig. 10), in agreement with the $Zn_4SO_4(OH)_6\cdot 4H_2O$ composition. The elemental mappings further suggest that the platelets formed at the discharged state are Zn-rich. The above results demonstrate that $Zn_4SO_4(OH)_6\cdot 4H_2O$ is formed upon discharge and its crystal water gives rise to the broad band in FT-IR. Since $Zn_4SO_4(OH)_6\cdot 4H_2O$ precipitation takes place in solutions with pH above 5.5[46,47], its formation in $ZnSO_4$ electrolyte is due to proton insertion into cathode which leaves $OH^-$ behind and causes local pH increase[19,48,49]. It suggests reversible proton de/insertion into TABQ during charge/discharge. Despite the insulating nature of $Zn_4SO_4(OH)_6\cdot 4H_2O$, it forms as individual platelets rather than uniform film over active material. Therefore, the electrical conductivity of the electrode is not hindered as confirmed by electrochemical impedance spectroscopy (EIS, Supplementary Fig. 11). Its reversible formation also functions as a pH buffer for the system.

Since $Zn^{2+}$ would also function as the charge carrier in the $ZnSO_4$ electrolyte, the amount of proton vs. $Zn^{2+}$ storage in TABQ was quantified by inductively coupled plasma optical emission spectroscopy (ICP-OES) and ion chromatography (IC). They result in the Zn and sulfate weight percentages of 17.34% and 5.76% in the discharged electrode, respectively, and the inserted proton is calculated to be 13.5 times of $Zn^{2+}$ (please find detailed calculations in Supplementary Discussions). It suggests the domination of proton storage in TABQ during the redox processes.

The dominated proton insertion in TABQ was further verified by the investigation of pH influence on cathode behaviors. The TABQ electrode was tested in three electrode cells with saturated calomel electrode (SCE) as the reference and a series of electrolytes: group A of pH~4 solutions, including 1 M $ZnSO_4$, 1 M Zn$(CF_3SO_3)_2$ and $5\times 10^{-5}$ M $H_2SO_4$; group B of pH~0 solutions, including 0.5 M $H_2SO_4$ and 1 M $ZnSO_4 + 0.5$ M $H_2SO_4$. The average charge/discharge potentials of TABQ in the pH~0 electrolytes (group B) are around 0.13 V vs. SCE, while the ones with the pH~4 electrolytes (group A) are around $-0.12$ V vs. SCE (Fig. 4c). Such pH dependent redox potential of TABQ

demonstrates proton involved reaction, and the potential difference obeys the Nernst equation (please find detailed calculations in Supplementary Discussions). The electrode presents high overpotential in the $5\times 10^{-5}$ M $H_2SO_4$ electrolyte due to the low ion activity, but the average potential is close to the other pH~4 electrolytes. In 1 M $ZnSO_4$ and $Zn(CF_3SO_3)_2$, on the other hand, continuous $Zn^{2+}$ hydrolysis provides enough proton for cathode reactions. The results confirm that proton is the dominated active cation associated with the redox of TABQ.

Importantly, the performance of zinc anode is not influenced by the proton de/insertion at the cathode, thanks to the reversible formation of $Zn_4(OH)_6SO\cdot 4H_2O$ as pH buffer. Supplementary Fig. 12 shows the potential curves of zinc electrode in a three-electrode cell with SCE reference and TABQ counter electrode. The flat and stable curves at different cycles demonstrate the excellent stability of Zn plating/stripping reaction in the system. It allows the coupling of zinc metal anode with a proton inserted cathode in the zinc-organic battery.

The unique proton insertion manner in TABQ should be attributed to its amino groups, which are available to protonate in the weakly acidic electrolyte of $ZnSO_4$. It allows continuous proton exchange with electrolyte to create a proton active environment, which initiates proton insertion and helps with desolvation. The protonation of TABQ was confirmed by UV–vis analysis. In the non-protonated TABQ, the lone pair electrons on the amino groups would form extended conjugation with the $\pi$ electrons on quinone rings[50,51]. Such interaction is disturbed upon the protonation of amino groups. Therefore, the K-band in protonated TABQ possesses similar energy with benzoquinone[52] and is blue-shifted in comparison to non-protonated TABQ due to the absence of extended conjugation. Figure 5a shows the UV-vis spectra of TABQ in water and 1 M $ZnSO_4$. A clear blueshift of K-band from 268 nm in water to 201 nm in 1 M $ZnSO_4$ is noted, confirming the protonation in the latter.

The effect of protonation on proton insertion is further extended to the compounds of 2,5-diamino-1,4-benzoquinone (DABQ)[37], THBQ and 2,5-dihydroxy-1,4-benzenediacetate (DOBDA)[53]. Supplementary Fig. 13a shows their charge-discharge curves, and the discharged electrodes were

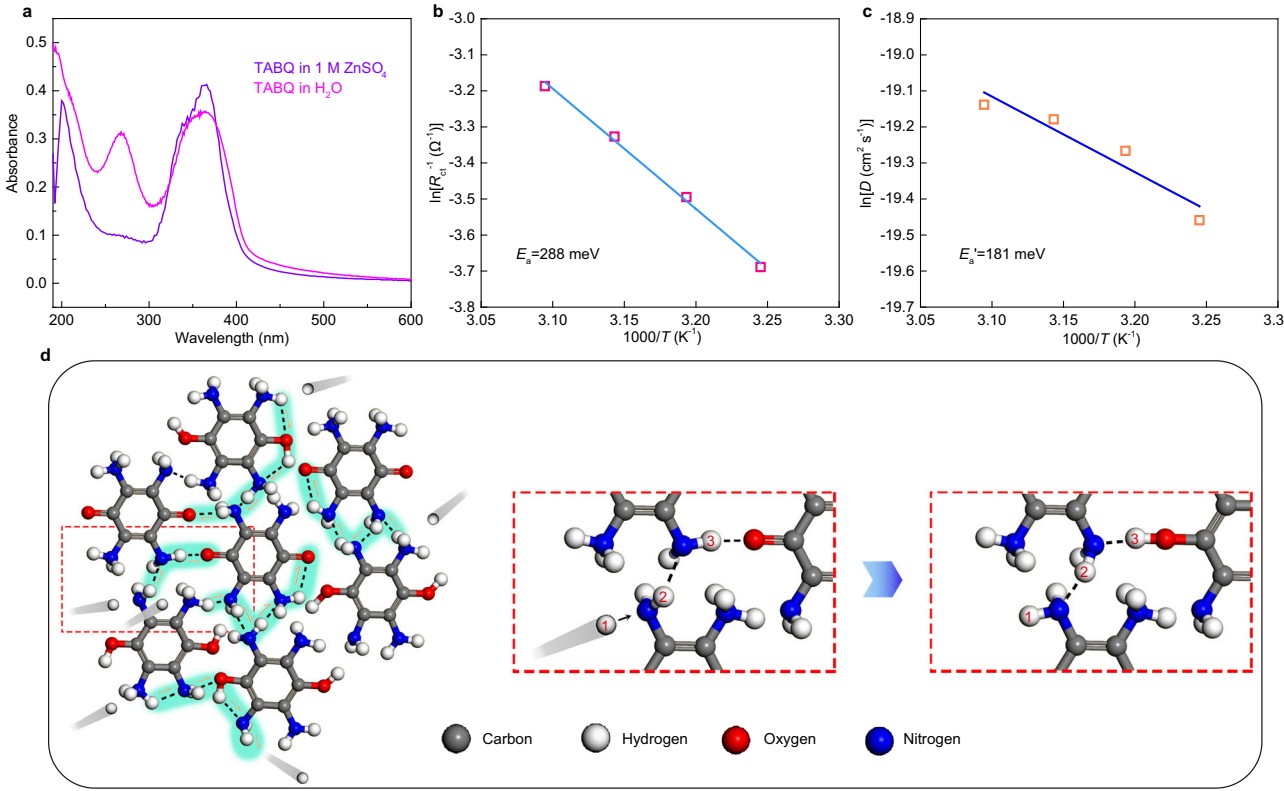

**Fig. 5 Proton insertion behavior into TABQ. a** UV–vis absorption spectra of TABQ in 1 M ZnSO₄ and water. The Arrhenius plots of **b** ln($R_{ct}$) vs. 1000/$T$ and **c** ln($D$) vs. 1000/$T$ for TABQ. **d** Schematic illustration for the proton conduction manner in the hydrogen bonding network in TABQ.

characterized by XRD (Supplementary Fig. 13b). The $Zn_4SO_4(OH)_6·4H_2O$ diffraction peak is shown in the discharged DABQ and DODBA, which indicates proton insertion. It is attributed to the active protonation/deprotonation of amino or carboxyl groups. On the other hand, the proton activity on hydroxyl groups is low in 1 M ZnSO₄[54], and the protonation of carbonyl groups on quinones is negligible[55–57]. Proton insertion is thus less favored in THBQ. Previously reported quinone materials undergoing Zn storage, such as TCBQ, PTO and C4Q[28,30,41], would not protonate in the weakly acidic zinc electrolytes, either.

The charge transfer behavior in TABQ was studied by EIS with measurements taken at various temperatures. A typical equivalent circuit was applied to calculate charge transfer resistance ($R_{ct}$) from the Nyquist plots (Supplementary Fig. 14 and Supplementary Table 1). A linear correlation was obtained between ln($R_{ct}^{-1}$) and $T^{-1}$ (Fig. 5b), suggesting the applicability of Arrhenius equation of ln($R_{ct}^{-1}$) = $-E_a/RT + C$[58,59]. The activation energy $E_a$ for charge transfer processes was calculated to be 288 meV. The low value suggests facile ion desolation thanks to the help of amino groups.

The proton conduction in TABQ was studied by galvanostatic intermittent titration technique (GITT) at various temperatures. The obtained diffusion coefficient ($D$) values were used to calculate the activation energy for ion diffusion according to the Arrhenius equation of ln($D$) = $-E_a'/RT + C'$ (Fig. 5c and Supplementary Fig. 15)[60]. It results in a low $E_a'$ value of 181 meV. Considering the hydrogen bonding network formed among TABQ molecules as discussed earlier, the result demonstrates a unique proton conduction manner via the Grotthuss-type mechanism ($E < 400$ meV)[61,62], where proton transfer takes place by forming/breaking of hydrogen bonds between adjacent carbonyl/hydroxyl and amino groups (Fig. 5d). Such process

allows much more facile ion conduction in comparison to the conventional diffusion mode. The charge/discharge kinetics of TABQ were studied by the method proposed by Dunn et al.[63]. The relationship of the peak current $i$ and scan rate $v$ from cyclic voltammetry (CV) obeys the equation of $i = av^b$. The $b$ values of the two pairs of TABQ redox peaks are close to 1 (Supplementary Fig. 16), suggesting the domination of non-diffusion-controlled process and indicating facile reaction kinetics[62,63]. It guarantees the excellent electrochemical performance of the Zn-TABQ battery.

## Discussion

In this work, specific functional groups are chosen for the small molecule quinone cathode of TABQ for aqueous zinc-organic batteries. The hydrogen bonds suppress the sublimation and provides convenience for various electrode treatments. The high molecular symmetry results in low solubility in aqueous electrolytes. More importantly, the redox of carbonyl group is associated with dominated proton de/insertion thanks to the amino groups which undergoes protonation in the weakly acidic zinc electrolyte. Facile proton conduction in TABQ is furthermore achieved with a Grotthuss-type mechanism through the hydrogen bonding network formed with the amino and carbonyl groups. Therefore, proton insertion in TABQ experiences much enhanced kinetics and is favored for energy storage. Despite the low concentration of proton in the electrolyte, the continuous hydrolysis of $Zn^{2+}$ generates sufficient proton for insertion. The unique proton conduction manner results in activation energies below 300 meV for charge transfer and proton diffusion, which guarantees a high capacity of 303 mAh g⁻¹ with excellent rate capability for the TABQ electrode. Stable cycling is also achieved for 1000 cycles. The unique proton conduction manner presented in our work

**Fig. 6 Synthetic routes of TABQ.** TABQ was obtained by a two-step synthesis with low-cost reactants.

proposes a promising direction for the design of high-performance organic electrode materials for aqueous batteries.

## Methods

**Materials.** Graphite foil was purchased from the SGL group (Germany). TCBQ, THBQ, o-TCBQ, DABQ, DOBDA, benzoquinone and potassium phthalimide were purchased from Aladdin Bio-Chem Technology (China). KB conductive additive was purchased from Lion Specialty Chemicals (Japan). Poly(vinylidene fluoride) (PVDF) was purchased from Kejing Materials Technology (China). The other reagents were obtained from Sinopharm Chemical Reagent (China).

**Synthesis of TABQ.** The TABQ was prepared according to a previously reported method[64] as summarized in Fig. 6 and described below.

Step 1: 20.0 g (0.081 mol) of TCBQ, 60.0 g (0.32 mol) of potassium phthalimide and 200.0 mL of acetonitrile were added into a 500 mL round bottom flask. The mixture was kept at 80 °C for 12 h under stirring. After cooling down to room temperature, the solid was separated by filtration and washed with 2 L boiled water. The product was dried at 60 °C overnight, and brown-yellow powder of tetra (phthalimido)-benzoquinone (TPQ) was obtained (45.0 g, yield = 80.7%).

Step 2: 9.41 g (0.014 mol) of TPQ was dispersed in 200 mL of 80% hydrazine hydrate in a 500 mL round bottom flask. The mixture was stirred at room temperature for 2 h, followed by heating at 65 °C for 2 h. The system was cooled down to room temperature, filtered, washed with water and ethanol, and the dark purple product of TABQ was obtained (1.3 g, yield = 56.6%). $^1$H NMR, (500 MHz, [D$_6$] DMSO): δ 4.55 (s, 8H) (Supplementary Fig. 17); elemental analysis (calcd., found for C$_6$H$_8$N$_4$O$_2$): C (42.86, 42.09), H (4.80, 4.03), N (33.32, 33.27), O (19.03, 21.41).

**Characterizations.** FT-IR was performed on VERTEX70 (Bruker, Germany). XRD was carried out on an Empyrean diffractometer with Cu-Kα radiation (PANalytical B.V., Holland). $^1$H NMR was measured on Bruker 500 M (Bruker, Germany) with (methyl sulfoxide)-d$_6$ as the solvent and tetramethylsilane (TMS) as the internal standard. Elemental analysis was determined by an Elementar Vario EL III (Elementar, Germany). TGA was carried out on a TGA/DSC3+ thermal analysis system (Mettler toledo, Switzerland). The morphology was obtained by a SU8010 SEM equipped with an EDS detector (HITACHI, Japan). UV–vis absorption spectra were recorded on a U-3900 spectrophotometer (HITACHI, Japan). XPS was carried out on a K-Alpha+ X-ray Photoelectron Spectroscopy (Thermo fisher Scientific, America). ICP-OES analysis was carried out on ICP-OES 730 (Agilent, America). IC analysis was carried out on ICS-1100 (DIONEX, America).

**Sublimation behavior study.** The benzoquinone or TABQ powder was placed on the aluminum foil cover of a long-neck round-bottom flask. It was placed upside down with the powder and neck sticking into the oven which was set at 90 °C or 150 °C and the round part staying outside at room temperature. Sublimation was evidenced by the disappearance of powder on the aluminum cover and condensation of crystals on the round part.

**UV–vis analysis.** Standard solutions were prepared by dissolving known compound concentrations in 1 M ZnSO$_4$. Saturated solutions were obtained by mixing an excess of compounds in 1 M ZnSO$_4$ and filtered. Saturated solutions were diluted by the factors of 100, 200 and 100 with 1 M ZnSO$_4$ for TABQ, THBQ and o-TCBQ, respectively. UV–vis absorbance was measured in the range of 190–600 nm. Calibration curves were obtained by the linear fit of the absorbance at 365 nm, 363 nm and 335 nm for TABQ, THBQ and o-TCBQ, respectively, with respect to the compound concentrations of standard solutions using the Beer-Lambert law: $A = \varepsilon lc$ ($A$: absorbance; $\varepsilon$: molar extinction coefficient; $l$: length of the cell; $c$: compound concentration).

**Electrochemical measurements.** The TABQ electrodes were prepared by mixing TABQ, KB conductive additive, and PVDF binder with a weight ratio of 5:4:1 in NMP. The slurry was drop casted on graphite foil discs with the diameter of 11

mm, and dried at 60 °C under vacuum overnight followed by 90 °C overnight. The mass loading of TABQ was around 1.3 mg cm$^{-2}$. Galvanostatic charge/discharge measurements were carried out in PFA Swagelok® type cells with titanium rods as the current collectors. Rest test was performed with the following process: regular discharge and charge for 20 cycles at 1 A g$^{-1}$, rest for 10 h after discharged to half capacity, fully discharged, charged to half capacity and fully charged, respectively (a total of 40 h rest period), and return to regular cycles. The study of electrolyte pH influence was carried out in T-shaped PFA Swagelok® type cells with TABQ working electrode, graphite foil counter electrode, SCE reference electrode and the aqueous electrolytes of 1 M ZnSO$_4$, 1 M Zn(CF$_3$SO$_3$)$_2$, 5 × 10$^{-5}$ M H$_2$SO$_4$, 0.5 M H$_2$SO$_4$, or 1 M ZnSO$_4$ + 0.5 M H$_2$SO$_4$. EIS measurements were carried out in a T-shaped PFA Swagelok® type cell with TABQ working electrode, Zn counter electrode, SCE reference electrode and 1 M ZnSO$_4$ aqueous electrolyte. GITT was carried out in a T-shaped PFA Swagelok® type cell with TABQ working electrode, Zn counter electrode, Zn reference electrode and 1 M ZnSO$_4$ aqueous electrolyte. The cells for EIS and GITT measurements were placed in an oven which was set to the desired temperatures of 35, 40, 45, and 50 °C. All electrochemical experiments were carried out on a Bio-Logic VMP3.

**In-situ XRD analysis.** The test was carried out with a home-made in-situ XRD cell. The TABQ slurry was drop casted on a carbon cloth substrate. The cell was discharged and charged to the desired voltages and XRD measurements were taken.

**Activation energy calculation.** The Nyquist plots from EIS were fitted with a typical equivalent circuit shown in Supplementary Fig. 14b, and the obtained $R_{ct}$ values were summarized in Supplementary Table 1. The low errors confirmed the applicability of the circuit for calculating $R_{ct}$. The $\ln(R_{ct}^{-1})$ values were plotted vs. $1000/T$ and linear fit was carried out according to the Arrhenius equation of $\ln(R_{ct}^{-1}) = -E_a/RT + C$, where $C$ is constant under a stable experimental condition, $R$ is gas constant and $T$ is temperature[62]. The $E_a$ represents the activation energy for charge transfer and was calculated from the slope of the fitted line. Similarly, the activation energy $E_a$' for diffusion was calculated from the Arrhenius equation with diffusion coefficient ($D$) of $\ln(D) = -E_a'/RT + C'$. $D$ was calculated from GITT based on the following equation:

$$D = \frac{4L^2}{\pi\tau}\left(\frac{\Delta E_s}{\Delta E_t}\right)^2 \tag{1}$$

where $\tau$ is the relaxation time, $\Delta E_s$ is the steady-state potential change after a single pulse, and $\Delta E_t$ is the potential change during a pulse after eliminating iR drop. The diffusion length $L$ was measured by the geometric thickness of cathode. Since $L$ was a constant, the value would not affect the activation energy obtained from slope of the Arrhenius equation. The linearity between cell voltage and $t^{1/2}$ during titration was checked to confirm the applicability of the equation (Supplementary Fig. 18)[65].

**Kinetics studies.** The charge/discharge kinetics of TABQ in aqueous zinc cells were studied by CV tests at various scan rates (Supplementary Fig. 16a). The CV curves showed two pairs of redox peaks. The peak current $i$ and scan rate $v$ obeys the relationship of $i = av^b$, where $a$ and $b$ are coefficients. In the limiting cases, a $b$ value of 0.5 suggests diffusion-controlled process whereas 1 indicates non-diffusion-controlled process[62,63]. The peak current at different scan rates was extracted, and the ln of peak current was plotted vs. ln of scan rate. By carrying out linear fit, the $b$ values were obtained from the slopes of the lines (Supplementary Fig. 16b).

## Data availability

The data that support the findings of this study are available from the corresponding author upon reasonable request.

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

## Acknowledgements

This work was supported by the National Natural Science Foundation of China (51974070), the LiaoNing Revitalization Talents Program (XLYC1907069), and the Fundamental Research Funds for the Central Universities (N2105001). Special thanks are due to the instrumental analysis from Analytical and Testing Center, Northeastern University and SDBSWeb: https://sdbs.db.aist.go.jp (National Institute of Advanced Industrial Science and Technology, 2020-09-01).

## Author contributions

Z.L. and X.S. conceived and designed this work. The experiment was performed by Z.L. under the guidance of X.S. and H.-Y.S. with the help of L.L. as well as X.Y.. W.W. carried out the SEM and XRD tests. All co-authors discussed the results. Z.L., X.S. and L.L. wrote the manuscript. All authors have given approval to the final version of the manuscript.

## Competing interests

The authors declare no competing interests.
