## [Peer Review File · Nature Communications]

REVIEWER COMMENTS

Reviewer #1 (Remarks to the Author):

In this manuscript, a small molecule quinone cathode, tetraamino-p-benzoquinone (TABQ), was used in zinc ion batteries. TABQ displays proton insertion in weak acidic zinc electrolyte. However, the energy mechanism is not discussed deeply. Furthermore, compared with previously reported work, the novelty of this work is not enough and this work does not provide significant academic information. Therefore, I consider for the rejection.

1. The capacity of TABQ in Figure 2a is not consistent with the result in Figure 2d at same current density.
2. The results show that the H⁺ will participate in the energy storage. More in-situ experimental characterizations such as XPS, solid state NMR and Raman have to be carried out at different charging/discharging states.
3. In order to illustrate the advantage of TABQ, authors could provide a systematic comparison between the performance of their work and that of previously reported similar materials.
4. More experimental details have to be included in the experimental section.
5. The charge/discharge kinetics of TABQ could be included and discussed deeply
6. Some important references are not cited.

Reviewer #2 (Remarks to the Author):

The authors reported quinone based molecular compound as a cathode material for aqueous zinc ion batteries (AZIBs). High capacity (300 mAh g⁻¹ at 0.1A g⁻¹), good rate capability (213 mA g⁻¹ at 5 A g⁻¹), and long cycle stability (> 85% capacity retention after 1000 cycles) have been demonstrated. Proton coordination as the species associated with the redox of TABQs has been proposed. Compared to previous research, proton storage shown in this work is potentially significant and publishable, but more evidence and discussion should be provided.

1. Proton storage in quinone molecules has been reported previously, and should be cited here, e.g. *Electrochim. Acta* 17, 873-887 (1972); *Nature Mater* 16, 841-848 (2017).
2. More evidence is needed to support the proton storage in TABQ because this is non-intuitive due to higher Zn²⁺ concentration than that of protons. Current EDS results are indirect. For example, the authors can provide quantitative analysis of EDS mapping of charged/discharged samples with spatial resolutions. Please refer to Fig 1C in *Angew. Chem. Int. Ed.* 2018, 57, 11737 and Fig S6 in *Chem. Mater.* 2018, 30, 3874-3881.
3. Previous work (*Angew. Chem. Int. Ed.* 2018, 57, 11737 and *Chem. Mater.* 2018, 30, 3874-3881) reported Zn²⁺ storage in similar quinones. If proton storage is indeed true in TABQ, as suggested by the authors, it is important for the authors to discuss the mechanisms behinds the seemingly conflicting storage modes (Zn²⁺ vs. H⁺ storage) for quinone molecules and reconcile the difference.
4. The authors claimed TABQ is insoluble in the aqueous electrolyte with excellent cycle life. It is the total cycling time rather than the number of cycles matters. The authors need to show stable cycling at 0.5C for at least 100 cycles.
5. The rest tests were performed only at fully charge/discharge states. How about taking rest test at 50% DOD?

Reviewer #3 (Remarks to the Author):

The authors report an interesting, valuable, and with potential follow up work on using tetraamino-p-benzoquinone (TABQ) as cathode material for aqueous Zn batteries. The new finding of intramolecular H-bonding induced anomalous yet facile proton conduction through Grotthus – type mechanism is original and of high relevance for the community. The materials/electrode/cells display high capacity of 303 mAh g⁻¹ with good rate capability and stable cycling. Data presentation and analysis is also of good quality so I can recommend this work for publication following these revision points are addressed:

- one point that worth being further discussed is the solubility of the studied material, and whether this is associated to H-bonding or not. The anomalous H-bonding material studied by authors case, reminds me of one paper I saw few years ago on intermolecular H-bonding in DABQ for solid Li-cells (DOI: 10.1039/C8SC02995D); wherein similar studies have been performed – like for example comparison with quinone reference molecule. Interesting is that in both cases, the studies are the results of curious behavior of these molecules, so I would suggest an extended discussion on this in the introductory part. And how about then using other organic chemistries, that would potentially allow also for such H-interactions, for example the recently developed high voltage organic materials for Li-ion batteries (DOI : 10.1038/s41563-020-00869-1 ; DOI : org/10.1021/acs.chemmater.0c02989 ; DOI : org/10.1016/j.jpowsour.2020.228814), but also applicable to other monovalent and divalent cations. Would these be suitable for the developed concept in this work. Again, a broader context discussion with also in the context and mention of these papers could broaden the view this work has on the field.
- The XRD pattern of TABQ shown in Fig. S1 displays amorphous or a poorly-crystalline phase. I would have expected TABQ to be very well-crystallized, in particular due to the rich intermolecular hydrogen bonding, highly symmetrical and small molecule. Do the authors specifically pursue this state, and if yes, how is this attained? or could the authors explain?
- also, in line 143, it is mentioned that XRD after 1000 cycles verifies the excellent structural stability. But according to my comment above, how this can be confirmed. Because if I compare the XRD data of the pristine TABQ (Fig. S1) and the one after 1000 cycles, besides the similar low-crystallinity, they are also quite different. In pristine, there are two peaks at 20° and 28°, while after cycling, the peaks at 9° and 27° become pronounced. I would suggest authors to double-check again the XRD data, as for now this cannot support the claim of excellent structural stability.
- TABQ is mentioned at the same time to be soluble in water, while also mentioning its insolubility in the electrolyte (1 M ZnSO₄ aqueous electrolyte) in Line 152. Salt concentration can impact the solubility of organics, but not sure it can render completely insoluble. Quantified data would be better to have so if possible, please provide proofs of solubility in the electrolyte, by immersing the compound directly in the electrolyte, eventually by measuring and quantifying this as in DOI: 10.1039/C8SC02995D.
- For electrochemical measurements, the authors used 40% of KB conductive agent which is a capacitance carbon, capable of provide large amount of capacity depending on the cycling rate. Since it is claimed that TABQ is not soluble in the electrolyte, the authors, if possible, should provide a charge-discharge data using other conductive carbon like Super P or similar? Or a pure KB electrode charge-discharge data can also make this work more rigorous.
- Line 198, the formation of Zn₄SO₄(OH)₆·4H₂O is due to the local pH increase. Why there is local pH increase, and why local pH increase will give rise to the formation of Zn₄SO₄(OH)₆·4H₂O. Could the authors give more detailed explanation/discussion in the paper, for example a citation of a previous work, so even a reader from different field can understand and follow easily this work?
- the pH dependent redox potential of TABQ is also slightly ambiguous. In the electrolyte of 1M ZnSO₄ (pH=4), the tetraamine is not able to be protonated to NH₃⁺, while in the 0.5M H₂SO₄ electrolyte, the tetraamine is fully protonated to NH₃⁺. In the case of the latter, NH₃⁺ serves as strong electron withdrawing group, decreasing the electron density of the redox center, and thus should increase the redox potential. So what plausible explanation can be provided for their data by the authors?

Dr. Xiaoqi Sun.

Tel: 86-24-83689510

Fax: 86-24-83684323

Email: sunxiaoqi@mail.neu.edu.cn

Response to Reviewers' Comments (Manuscript number NCOMMS-21-00810)

We thank the reviewers very much for their careful review of our manuscript. We have addressed or clarified the issues raised in the reviewers' reports. Our responses are as follows.

Response to reviewer 1

Comments:

Reviewer 1: In this manuscript, a small molecule quinone cathode, tetraamino-p-benzoquinone (TABQ), was used in zinc ion batteries. TABQ displays proton insertion in weak acidic zinc electrolyte. However, the energy mechanism is not discussed deeply. Furthermore, compared with previously reported work, the novelty of this work is not enough and this work does not provide significant academic information. Therefore, I consider for the rejection.

Response:

We thank the reviewer for the careful review of our manuscript and the insightful comments. We have added in-situ and ex-situ characterizations to further study the energy storage mechanism of TABQ. Please find the details in our response to question #2. Regarding the novelty of the work, we are sorry that we may not have discussed it deeply enough in the original manuscript. We have added more experiments and discussions to further demonstrate the novelty. The major point is that a unique Grotthuss-type proton conduction manner is revealed for TABQ, which has never been reported for quinone materials in aqueous zinc cells. This conduction is realized by the hydrogen bonding network formed among amino and carbonyl groups on TABQ molecules and allows excellent reaction kinetics. Specifically, TABQ shows the best rate capability among quinone cathode materials in aqueous zinc batteries (please find the detailed comparison in our response to question #3).

The origin of proton insertion in TABQ is further studied in the revised manuscript. The molecule contains four amino groups that are available for protonation in the weakly acidic electrolyte of 1 M ZnSO₄ with pH around 4. Therefore, the amino groups experience continuous proton exchange with

electrolyte to create a proton active environment around TABQ, which initiates the proton insertion process. The protonation can be characterized by UV-vis spectroscopy. In a non-protonated TABQ molecule, the lone pair electrons on the amino groups form extended conjugation with the π electrons on the quinone ring [J. Mol. Struct. 1984, 114, 249-252; J. Am. Chem. Soc. 2002, 124, 2518–2527]. This extended conjugation is disturbed upon the protonation of amino groups. Therefore, the B-band of quinone ring in protonated TABQ possesses similar energy with benzoquinone [Energy Environ. Sci. 2017, 10, 2372] and is blue-shifted in comparison to non-protonated TABQ due to the absence of extended conjugation. Figure R1a shows the UV-vis spectra of TABQ in water and 1 M ZnSO_4 . A clear blueshift of B-band from 268 nm in water to 201 nm in 1 M ZnSO_4 is noticed, confirming its protonation in the latter. The proton insertion into TABQ is thus initiated.

The effect of protonation on proton insertion can be further extended to other molecules. Charge-discharge tests were performed on the quinone cathodes of 2,5-diamino-1,4-benzoquinone (DABQ), tetrahydroxy-p-benzoquinone (THBQ) and 2,5-dihydroxy-1,4-benzenediacetate (DOBDA) in 1 M ZnSO_4 , and XRD measurements of discharged electrodes were carried out (Figure R1b, c). The discharged DABQ and DOBDA show diffractions from $\text{Zn}_4\text{SO}_4(\text{OH})_6 \cdot 4\text{H}_2\text{O}$. Notably, $\text{Zn}_4\text{SO}_4(\text{OH})_6 \cdot 4\text{H}_2\text{O}$ precipitates in solutions with pH value above 5.5 and dissolves under lower pH environment [ChemSusChem, 2016, 9, 2948-2956]. Its formation in zinc cells results from local pH increase along with proton insertion into the cathode during discharge and is widely used as the indicator for proton involved reactions [Nat. Energy 2016, 1, 16039; Nat. Commun. 2018, 9, 1656; Nat. Commun. 2018, 9, 2906]. Therefore, proton insertion takes place in DABQ and DOBDA. It is attributed to the active proton exchange between electrolyte and amino or carboxyl groups, respectively. On the other hand, the hydroxyl groups on THBQ cannot protonate or deprotonate in 1 M ZnSO_4 so that proton insertion is not observed. Previously reported quinone molecules, such as calix[4]quinone (C4Q), pyrene-4,5,9,10-tetraone (PTO) and tetrachloro-p-benzoquinone (TCBQ) [Sci. Adv. 2018, 4, eaao1761; Angew. Chem. Int. Ed. 2018, 57, 11737-11741; Chem. Mater. 2018, 30, 3874-3881], do not have protonation sites on the molecules either and Zn^{2+} insertion takes place instead.

Figure R1 | The studies of proton insertion in quinone cathode materials. a UV-vis absorption spectra of TABQ in 1 M ZnSO_4 and water. **b** Charge/discharge curves of DOBDA, THBQ, DABQ and **c** XRD patterns of the discharged electrodes.

Overall, it is revealed that the protonation of quinone molecules is the initiator for proton insertion, and hydrogen bonding network formed among molecules ensures facile proton conduction through Grotthuss-type mechanism. Our work proposes a strategy to design high performance electrode materials. We have added the new results and discussions to the revised manuscript (Page 15-16, 19, Figure 5a in the revised manuscript and Supplementary Figure 13 in the revised supplementary information). Our point-to-point responses to other questions are listed below.

1. *The capacity of TABQ in Figure 2a is not consistent with the result in Figure 2d at same current density.*

Response:

We thank the reviewer for the comment. The TABQ electrode undergoes slight capacity decay during early cycles followed by stabilization upon later cycles. The voltage profiles drawn in Figure 2a are from the stabilized cycles. Therefore, the capacity is not the same as the initial cycle of the cycling test at 5 A g⁻¹ in Figure 2d. We have added the discussion in the revised manuscript (Page 9 in the revised manuscript).

2. *The results show that the H⁺ will participate in the energy storage. More in-situ experimental characterizations such as XPS, solid state NMR and Raman have to be carried out at different charging/discharging states.*

Response:

We thank the reviewer for the suggestion. We carried out in-situ XRD test to study the cation insertion behavior (Figure R2a,b). During discharge, a peak at around 8.6 degrees shows up, which corresponds to the Zn₄SO₄(OH)₆·4H₂O phase. The peak disappears upon re-charge. As demonstrated in literature, the formation of Zn₄SO₄(OH)₆·4H₂O during discharge in ZnSO₄ aqueous electrolytes results from proton insertion into cathode materials which leaves OH⁻ behind and causes local pH increase. Zn₄SO₄(OH)₆·4H₂O precipitates from ZnSO₄ solutions when pH rises above 5.5 [ChemSusChem 2016, 9, 2948-2956; Electrochim. Acta 2012, 85, 438-443]. When proton de-inserts from cathode materials upon charge, local pH decreases and Zn₄SO₄(OH)₆·4H₂O dissolves back to the electrolyte. The reversible formation of Zn₄SO₄(OH)₆·4H₂O is widely used as an indicator for proton involved energy storage processes in aqueous zinc batteries [Nat. Energy 2016, 1, 16039; Nat. Commun. 2018, 9, 1656; Nat. Commun. 2018, 9, 2906]. Our in-situ XRD results show the reversible formation of Zn₄SO₄(OH)₆·4H₂O at the cathode and confirm the reversible proton de-insertion into TABQ. The reversible formation of Zn₄SO₄(OH)₆·4H₂O is furthermore demonstrated by EDS mappings (Figure R2c). We also carried out in-situ Raman analysis as suggested by the reviewer. However, water in electrolyte as well as crystal water in Zn₄SO₄(OH)₆·4H₂O covers the O-H stretching from discharged

TABQ. Similar problems can be expected for XPS analysis, and we were not able to get access to in-situ solid state NMR. Nevertheless, the in-situ XRD results, together with the elemental characterizations and analysis of pH influence on redox potential, demonstrate the reversible proton de-insertion into TABQ. It contributes to the low activation energies for charge transfer and ion diffusion. We have added the results to the revised manuscript to further prove the participation of H^+ in energy storage (Page 12, 13, Figure 4b in the revised manuscript and Supplementary Figure 9 in the revised supplementary information).

Figure R2 | H^+ participation studies. **a** Schematic diagram of in-situ XRD analysis. **b** XRD peak evolution along discharge and charge. **c** SEM images and EDS mappings of TABQ at different states.

3. In order to illustrate the advantage of TABQ, authors could provide a systematic comparison between the performance of their work and that of previously reported similar materials.

Response:

We thank the reviewer for the suggestion. We compared the electrochemical performance of TABQ with previously reported quinone cathode materials for zinc-organic batteries, including calix[4]quinone (C4Q), pyrene-4,5,9,10-tetraone (PTO), HqTp, poly(benzoquinonyl sulfide) (PBQS), triangular phenanthrenequinone-based macrocycle (PQ- Δ), dibenzo[b,i]thianthrene-5,7,12,14-tetraone (DTT) and tetrachloro-p-benzoquinone (TCBQ) [Sci. Adv. 2018, 4, eaao1761; Angew. Chem. Int. Ed. 2018, 57, 11737-11741; Chem. Sci. 2019, 10, 8889-8894; Inorg. Chem. Front. 2018, 5, 1391-1396; J. Am. Chem. Soc. 2020, 142, 2541–2548; Adv. Mater. 2020, 2000338; Chem. Mater. 2018, 30, 3874-3881]. As shown in Figure R3, TABQ delivers the highest capacity of 303 mAh g⁻¹ at 0.1 A g⁻¹. With the increase of current density to 5 A g⁻¹, it also provides the highest capacity of 213 mAh g⁻¹ and demonstrates the capacity retention of 70% with 50 times increase of current density. The superior performance of TABQ is attributed to the unique proton conduction mechanism. We have added the discussion to the revised manuscript (Page 9 and Figure 2d in the revised manuscript).

Figure R3 | Electrochemical performance comparison of TABQ with previously reported quinone compounds (C4Q, PTO, HqTp, PBQS, PQ- Δ , DTT and TCBQ) in aqueous zinc-organic batteries.

4. *More experimental details have to be included in the experimental section.*

Response:

We thank the reviewer for the suggestion. The detailed synthesis procedure of TABQ has been added to the experimental section in the revised manuscript (Page 19-20 and Scheme 1 in the revised manuscript), and is listed below.

The TABQ was prepared based on the reactions shown in Scheme R1.

Step 1: 20.0 g of tetrachloride-p-benzoquinone (TCBQ, 0.081 mol), 60.0 g of potassium phthalimide (0.32 mol) and 200.0 mL of acetonitrile were added into a 500 mL round bottom flask. The mixture was kept at 80 °C for 12 h under stirring. After cooling down to room temperature, the solid was separated by filtration and washed with 2 L boiled water. The product was dried at 60 °C overnight, and brown-yellow powder of tetra(phthalimido)-benzoquinone (TPQ) was obtained (45.0 g, yield = 80.7%).

Step 2: 9.41 g (0.014 mol) of TPQ was dispersed in 200 mL of 80% hydrazine hydrate in a 500 mL round bottom flask. The mixture was stirred at room temperature for 2 h, followed by heating at 65 °C for 2 h. The system was cooled down to room temperature, filtered, washed with ethanol, and the dark purple product of TABQ was obtained (1.3 g, yield = 56.6%).

Scheme R1 | Synthetic routes of TABQ.

5. *The charge/discharge kinetics of TABQ could be included and discussed deeply.*

Response:

We thank the reviewer for the suggestion. The charge/discharge kinetics of TABQ were studied by cyclic voltammetry (CV) tests at various scan rates (Figure R4a). The CV curves contain two pairs of redox peaks. The peak current i and scan rate v obeys the relationship of $i=av^b$, where a is a coefficient and b describes the kinetics process. In the limiting cases, a b value of 0.5 suggests diffusion-controlled process whereas 1 indicates non-diffusion-controlled process [J. Phys. Chem. C

2007, 111, 14925-14931; Nat. Energy 2019, 4, 123-130]. The peak current at different scan rates was extracted, and the \ln of peak current was plotted vs. \ln of scan rate. By carrying out linear fit, the b values were obtained from the slopes of the lines (Figure R4b). The b values of the two pairs of redox peaks are close to 1, suggesting the domination of non-diffusion-controlled process and indicating facile reaction kinetics. We have added the discussion to the revised manuscript (Page 18 and 23-24 in the revised manuscript and Supplementary Figure 16 in the revised supplementary information).

Figure R4 | Kinetics study of TABQ. a CV curves at various scan rates. **b** The linear fits of $\ln(i)$ vs. $\ln(v)$ plots to calculate b values according to the equation of $i = av^b$.

6. *Some important references are not cited.*

Response:

We thank the reviewer for the comment. We have added the important references to the revised manuscript, including Reference 37 as an example of hydrogen bonding network constructed by amino groups [Chem. Sci. 2019, 10, 418-426]; Reference 34 to 36 and 45 for redox reactions of quinone materials [Electrochim. Acta 1972, 17, 873-887; Nature 2020, 579, 224-228; Nat. Mater. 2020, 10.1038/s41563-020-00869-1; Nat. Mater. 2017, 16, 841-848]; Reference 46 to 49 for the formation mechanism of Zn₄SO₄(OH)₆·4H₂O [Electrochim. Acta 2012, 85, 438-443; ChemSusChem 2016, 9, 2948-2956; Nat. Commun. 2018, 9, 1656 and Nat. Commun. 2018, 9, 2906].

Response to reviewer 2

Comments:

Reviewer 2: The authors reported quinone based molecular compound as a cathode material for aqueous zinc ion batteries (AZIBs). High capacity (300 mAh g⁻¹ at 0.1 A g⁻¹), good rate capability (213 mA g⁻¹ at 5 A g⁻¹), and long cycle stability (> 85% capacity retention after 1000 cycles) have been demonstrated. Proton coordination as the species associated with the redox of TABQs has been proposed. Compared to previous research, proton storage shown in this work is potentially significant and publishable, but more evidence and discussion should be provided.

Response:

We thank the reviewer for the positive comments and helpful suggestions to improve our manuscript. We have added EDS mapping to further confirm the proton insertion process (please find the details in our response to question #2). We have also added more experiments to reveal the origin of proton insertion into TABQ and discussed the proton vs. Zn²⁺ insertion into cathode materials of zinc batteries (please find the details in our response to question #3). Our point-to-point response and modifications of the manuscript are listed below.

1. *Proton storage in quinone molecules has been reported previously, and should be cited here, e.g. Electrochim. Acta 17, 873-887 (1972); Nature Mater 16, 841–848 (2017).*

Response:

We thank the reviewer for the advice. Those important references have been cited in the introduction part of the revised manuscript (Page 4 and Reference 34, 36).

2. *More evidence is needed to support the proton storage in TABQ because this is non-intuitive due to higher Zn²⁺ concentration than that of protons. Current EDS results are indirect. For example, the authors can provide quantitative analysis of EDS mapping of charged/discharged samples with spatial resolutions. Please refer to Fig 1C in Angew. Chem. Int. Ed. 2018, 57, 11737 and Fig S6 in Chem. Mater. 2018, 30, 3874–3881.*

Response:

We thank the reviewer for the suggestion. We carried out EDS elemental mappings on the pristine, discharged and charged TABQ cathodes (Figure R5). It shows Zn-rich platelets on TABQ particles in the discharged electrode, suggesting they are composed of Zn₄SO₄(OH)₆·4H₂O. As demonstrated in literature, the formation of Zn₄SO₄(OH)₆·4H₂O platelets during discharge in ZnSO₄ aqueous electrolytes results from proton insertion into cathode materials which leaves OH⁻ behind

and causes local pH increase. $Zn_4SO_4(OH)_6 \cdot 4H_2O$ precipitates from $ZnSO_4$ solutions when pH rises above 5.5 [ChemSusChem 2016, 9, 2948-2956; Electrochim. Acta 2012, 85, 438-443]. When proton de-inserts from cathode materials upon charge, local pH decreases and $Zn_4SO_4(OH)_6 \cdot 4H_2O$ dissolves back to the electrolyte. The reversible formation of $Zn_4SO_4(OH)_6 \cdot 4H_2O$ is widely used as an indicator for proton involved energy storage processes in aqueous zinc batteries [Nat. Energy 2016, 1, 16039; Nat. Commun. 2018, 9, 1656; Nat. Commun. 2018, 9, 2906]. The EDS mappings confirm the reversible proton de-insertion into TABQ during charge and discharge. We agree with the reviewer that the concentration of proton is lower than Zn^{2+} in the electrolyte. Nevertheless, the hydrolysis of Zn^{2+} continuously generates proton which makes it sufficient for insertion. We have added the discussion to the revised manuscript (Page 13, 19 and Figure 4b in the revised manuscript).

Figure R5 | SEM images and EDS mappings of the TABQ cathode at different stages.

3. *Previous work (Angew. Chem. Int. Ed. 2018, 57, 11737 and Chem. Mater. 2018, 30, 3874–3881) reported Zn^{2+} storage in similar quinones. If proton storage is indeed true in TABQ, as suggested by the authors, it is important for the authors to discuss the mechanisms behinds the seemingly conflicting storage modes (Zn^{2+} vs. H^+ storage) for quinone molecules and reconcile the difference.*

Response:

We thank the reviewer for the suggestion. Indeed, most of the previously reported quinone materials experience Zn^{2+} storage [TCBQ, Chem. Mater. 2018, 30, 3874-3881; PTO, Angew. Chem. Int. Ed. 2018, 57, 11737-11741; C4Q, Sci. Adv. 2018, 4, eaao1761; etc.]. The unique proton storage behavior with TABQ should be attributed to its special functional groups in comparison to those previously studied quinones. Firstly, the TABQ molecule contains four amino groups that are available

for protonation in the weakly acidic electrolyte of 1 M ZnSO₄ with pH around 4. Therefore, the amino groups experience continuous proton exchange with electrolyte to create a proton active environment around TABQ, which initiates the proton insertion process. The protonation of amino groups in TABQ can be characterized by UV-vis spectroscopy. In a non-protonated TABQ molecule, the lone pair electrons on the amino groups form extended conjugation with the π electrons on the quinone ring [J. Mol. Struct. 1984, 114, 249-252; J. Am. Chem. Soc. 2002, 124, 2518–2527]. This extended conjugation is disturbed upon the protonation of amino groups. Therefore, the B-band of quinone ring in protonated TABQ possesses similar energy with benzoquinone [Energy Environ. Sci. 2017, 10, 2372] and is blue-shifted in comparison to non-protonated TABQ due to the absence of extended conjugation. Figure R6a shows the UV-vis spectra of TABQ in water and 1 M ZnSO₄. A clear blueshift of B-band from 268 nm in water to 201 nm in 1 M ZnSO₄ is noticed, confirming its protonation in the latter. Proton insertion into TABQ is thus initiated. In comparison, previously reported quinone materials such as TCBQ, PTO and C4Q do not contain protonation sites and proton insertion is not favored.

The effect of protonation on proton insertion is further extended to other molecules. Charge-discharge tests were performed on the quinone cathodes of 2,5-diamino-1,4-benzoquinone (DABQ), tetrahydroxy-p-benzoquinone (THBQ) and 2,5-dihydroxy-1,4-benzenediacetate (DOBDA) in 1 M ZnSO₄, and XRD measurements of the discharged electrodes were carried out (Figure R6b, c). The formation of Zn₄SO₄(OH)₆·4H₂O is used as an indicator for proton insertion from ZnSO₄ electrolytes according to previous reports [Nat. Energy 2016, 1, 16039; Nat. Commun. 2018, 9, 1656; Nat. Commun. 2018, 9, 2906]. The Zn₄SO₄(OH)₆·4H₂O diffraction is formed in the discharged DABQ and DOBDA electrodes, suggesting proton insertion reactions. It is attributed to the active proton exchange between electrolyte and amino or carboxyl groups, respectively. On the other hand, the hydroxyl groups on THBQ cannot protonate or deprotonate in 1 M ZnSO₄ so that proton insertion is not observed.

Figure R6 | The studies of proton insertion in quinone cathode materials. a UV-vis absorption spectra of TABQ in 1 M ZnSO₄ and water. **b** Charge/discharge curves of DOBDA, THBQ, DABQ and **c** XRD patterns of the discharged electrodes.

In addition, hydrogen bonding network is formed among TABQ molecules with the amino and carbonyl functional groups as confirmed by FT-IR analysis in the original manuscript [Chem. Sci. 2019,

10, 418-426]. It creates a facile proton conduction path through the Grotthuss-type mechanism. Zn^{2+} diffusion, on the other hand, could be much more sluggish especially with the divalent charge and is less favored. Despite the lower concentration of proton than Zn^{2+} in the electrolyte, the continuous hydrolysis of Zn^{2+} generates sufficient proton for insertion. The above factors are the major causes of proton insertion over zinc insertion in TABQ. We have added the discussion to the revised manuscript (Page 4, 15-16, 19, Figure 5a in the revised manuscript and Supplementary Figure 13 in the revised supplementary information).

4. *The authors claimed TABQ is insoluble in the aqueous electrolyte with excellent cycle life. It is the total cycling time rather than the number of cycles matters. The authors need to show stable cycling at 0.5C for at least 100 cycles.*

Response:

We thank the reviewer for the suggestion. The long-term cycling was carried out at the low current density of 0.16 A g^{-1} , which is around 0.5 C. The TABQ electrode maintains 70% of the initial capacity after 100 cycles, and the total cycling time is around 300 hours (Figure R7a). The solubility of TABQ in 1 M ZnSO_4 was quantified by UV-vis analysis (Figure R7b) [Chem. Sci. 2019, 10, 418-426]. Standard solutions of TABQ in 1 M ZnSO_4 with concentrations of 10, 15, 20 and $45 \mu\text{mol L}^{-1}$ were measured in the range of 190-600 nm. The absorbance at 365 nm was plotted against concentration, and linear fits were carried out to obtain the calibration curve according to the Beer-Lambert law of $A=\epsilon lc$ (A : absorbance; ϵ : molar extinction coefficient; l : length of the cell; c : TABQ concentration). The saturated TABQ in 1 M ZnSO_4 was diluted 100 times with 1 M ZnSO_4 , and UV-vis measurement was carried out on the diluted solution. The concentration was calculated from the calibration curve. It results in the saturated concentration of 1.7 mmol L^{-1} TABQ in 1 M ZnSO_4 , suggesting a very low solubility of TABQ. It is attributed to the symmetric structure of the molecule which possesses low dipole moments and thus low solubilities in aqueous solutions [Chem. Sci. 2019, 10, 418-426]. The effect of dissolved TABQ on the Zn anode was further studied by cycling TABQ cathode and Zn anode in the electrolyte of saturated TABQ in 1 M ZnSO_4 . After 10 cycles at 0.1 A g^{-1} , the anode does not show any nitrogen signal according to X-ray photoelectron spectroscopy (XPS, Figure R7c). It suggests that the small amount of TABQ dissolved in electrolyte does not react with Zn, so that it does not cause continuous loss of active material or self-discharge. We have added the discussion to the revised manuscript (Page 5, 7-8, 11 and Figure 1f in the revised manuscript; Supplementary Figure 5 and Supplementary Figure 8 in the revised supplementary information).

Figure R7 | Cycling study and dissolution study of TABQ **a** Capacity and coulombic efficiency evolution at 0.5 C (0.16 A g⁻¹) for 100 cycles. **b** The UV-vis calibration curve of TABQ to calculate the saturated concentration. **c** XPS spectrum of Zn anode after 10 cycles at 0.1 A g⁻¹ in the electrolyte of 1 M ZnSO₄ with saturated TABQ.

5. The rest tests were performed only at fully charge/discharge states. How about taking rest test at 50% DOD?

Response:

We thank the reviewer for the suggestion. A new rest test was carried out, with the battery rested for 10 h after discharged to half capacity, fully discharged, charged to half capacity and fully charged, respectively. It gives a total rest period of 40 h (Figure R8a). The capacities before, during and after the rest cycles are the same (Figure R8b-d), verifying the excellent stability of TABQ cathode at various states. We have added the discussion to the manuscript (Page 10-11 in the revised manuscript and Supplementary Figure 7 in the revised supplementary information).

Figure R8 | Electrochemical performance of the TABQ cathode during rest test. a Voltage evolution over time. **b** Capacity evolution of TABQ with the rest cycle. **c** Charge/discharge curves and **d** differential capacity curves before, during and after the cycle with rest.

Response to reviewer 3

Comments:

Reviewer 3: The authors report an interesting, valuable, and with potential follow up work on using tetraamino-p-benzoquinone (TABQ) as cathode material for aqueous Zn batteries. The new finding of intramolecular H-bonding induced anomalous yet facile proton conduction through Grotthuss – type mechanism is original and of high relevance for the community. The materials/electrode/cells display high capacity of 303 mAh g⁻¹ with good rate capability and stable cycling. Data presentation and analysis is also of good quality so I can recommend this work for publication following these revision points are addressed:

Response:

We thank the reviewer for the positive comments and helpful suggestions for our manuscript. We have further revised the manuscript according to the reviewer's suggestions. Please find the details below.

1. *One point that worth being further discussed is the solubility of the studied material, and whether this is associated to H-bonding or not. The anomalous H-bonding material studied by authors case, reminds me of one paper I saw few years ago on intermolecular H-bonding in DABQ for solid Li-cells (DOI: 10.1039/C8SC02995D); wherein similar studies have been performed – like for example comparison with quinone reference molecule. Interesting is that in both cases, the studies are the results of curious behavior of these molecules, so I would suggest an extended discussion on this in the introductory part. And how about then using other organic chemistries, that would potentially allow also for such H-interactions, for example the recently developed high voltage organic materials for Li-ion batteries (DOI : 10.1038/s41563-020-00869-1 ; DOI : org/10.1021/acs.chemmater.0c02989 ; DOI : org/10.1016/j.jpowsour.2020.228814), but also applicable to other monovalent and divalent cations. Would these be suitable for the developed concept in this work. Again, a broader context discussion with also in the context and mention of these papers could broaden the view this work has on the field.*

Response:

We thank the reviewer for the helpful suggestion. We quantified the solubility of TABQ and compared it with the related compounds of tetrahydroxy-p-benzoquinone (THBQ) and tetrachloro-o-benzoquinone (o-TCBQ) by UV-vis analysis [Chem. Sci. 2019, 10, 418-426]. Specifically, the calibration curves of those compounds were made by dissolving known concentrations in 1 M ZnSO₄ and carrying out linear fit between absorbance and concentration according to the Beer-Lambert law of $A = \epsilon lc$ (A : absorbance; ϵ : molar extinction coefficient; l : length of the cell; c : concentration). Their saturated

solutions were diluted by the factors of 100, 200 and 100 with 1 M ZnSO₄ for TABQ, THBQ and o-TCBQ, respectively. The UV-vis absorbance was measured and the concentrations were calculated from the calibration curves. It results in the saturated concentrations of 1.7 mmol L⁻¹, 3.4 mmol L⁻¹ and 2.3 mmol L⁻¹ for TABQ, THBQ and o-TCBQ, respectively (Figure R9). In addition, literature has demonstrated the insoluble nature of tetrachloro-p-benzoquinone (p-TCBQ) [Chem. Mater. 2018, 30, 3874-3881]. We also observed trace dissolution of p-TCBQ, which makes it difficult to obtain an accurate concentration. Comparing among p-quinones, i.e. TABQ, THBQ and p-TCBQ, the higher solubilities of the former two compounds should be attributed to the more hydrophilic amino or hydroxyl functional groups. Although intermolecular H-bonding helps suppress dissolution in organic solutions as demonstrated in the paper mentioned by the reviewer [Chem. Sci. 2019, 10, 418-426], it would be less effective in aqueous solutions. Nevertheless, the paper also revealed that molecules with high symmetry possess low dipole moments and thus low solubilities. It would be the reason for the low solubilities of TABQ and THBQ. In accordance, the p-TCBQ molecule is more symmetric than o-TCBQ and presents lower solubility. We have added the above extended discussions and the important reference in the revised manuscript (Page 5, 7-8, Figure 1f, Reference 37 in the revised manuscript and Supplementary Figure 2 in the revised supplementary information)

Figure R9 | UV-vis calibration curves of a THBQ, b TABQ and c o-TCBQ in 1 M ZnSO₄.

The reviewer also brought up a few interesting molecules which help us to extend the concept of our work. We further studied the electrochemical behaviors of 2,5-diamino-1,4-benzoquinone (DABQ), THBQ and 2,5-dihydroxy-1,4-benzenediacetate (DOBDA) [Chem. Sci. 2019, 10, 418-426; Chem. Mater. 2020, 32, 23, 9996–10006], in order to investigate the origin of proton insertion. Figure R10a shows their charge-discharge curves. DABQ, THBQ and DOBDA deliver discharge capacities of 256, 164 and 128 mAh g⁻¹, respectively. The discharge electrodes were characterized by XRD (Figure R10b), and the Zn₄SO₄(OH)₆·4H₂O diffractions are shown in the discharged DABQ and DOBDA electrodes. Notably, Zn₄SO₄(OH)₆·4H₂O precipitates in solutions with pH value above 5.5 and dissolves under lower pH environment [ChemSusChem, 2016, 9, 2948-2956]. Its formation in zinc cells results from local pH increase along with proton insertion into the cathode during discharge and is widely used as the indicator for proton involved reactions [Nat. Energy 2016, 1, 16039; Nat. Commun. 2018, 9, 1656; Nat.

Commun. 2018, 9, 2906]. Therefore, proton insertion takes place in DABQ and DOBDA, as well as TABQ as demonstrated in the manuscript. The major character of those molecules is that TABQ and DABQ contains amino groups which can protonate in 1 M ZnSO₄ (pH ~ 4). It allows continuous proton exchange with electrolyte to create a proton active environment and initiate the proton insertion process. The carboxyl groups of DOBDA provide a similar proton active environment. The hydroxyl groups on THBQ, on the other hand, cannot protonate or deprotonate in 1 M ZnSO₄ and proton insertion is not observed. Zn storage has also been demonstrated for other quinone molecules that do not have protonation sites, such as calix[4]quinone (C4Q), pyrene-4,5,9,10-tetraone (PTO) and tetrachloro-p-benzoquinone (TCBQ) [Sci. Adv. 2018, 4, eaao1761; Angew. Chem. Int. Ed. 2018, 57, 11737-11741; Chem. Mater. 2018, 30, 3874-3881].

The protonation of TABQ in 1 M ZnSO₄ is further confirmed by UV-vis analysis. In a non-protonated TABQ molecule, the lone pair electrons on the amino groups form extended conjugation with the π electrons on the quinone ring [J. Mol. Struct. 1984, 114, 249-252; J. Am. Chem. Soc. 2002, 124, 2518–2527]. This extended conjugation is disturbed upon the protonation of amino groups. Therefore, the B-band of quinone ring in protonated TABQ possesses similar energy with benzoquinone [Energy Environ. Sci. 2017, 10, 2372] and is blue-shifted in comparison to non-protonated TABQ due to the absence of extended conjugation. Figure R10c shows the UV-vis spectra of TABQ in water and 1 M ZnSO₄. A clear blueshift of B-band from 268 nm in water to 201 nm in 1 M ZnSO₄ is noted, confirming its protonation in the latter.

We have added the above extended discussions on the proton insertion mechanism of organic molecules and the important references in the revised manuscript in order to broaden the view this work (Page 15-16, Figure 5a, Reference 37, 53 in the revised manuscript, Supplementary Figure 13 in revised supplementary information).

Figure R10 | Electrochemical studies of DOBDA, THBQ and DABQ in aqueous zinc cells and UV-vis spectra of TABQ. a Charge/discharge curves of DOBDA, THBQ, DABQ and **b** XRD patterns of the discharged electrodes. **c** UV-vis absorption spectra of TABQ in 1 M ZnSO₄ and water.

2. The XRD pattern of TABQ shown in Fig. S1 displays amorphous or a poorly-crystalline phase. I would have expected TABQ to be very well-crystallized, in particular due to the rich intermolecular hydrogen bonding, highly symmetrical and small molecule. Do the authors specifically pursue this state, and if yes, how is this attained? or could the authors explain?

Response:

We thank the reviewer for the question. As the reviewer expected, the as-prepared TABQ is highly crystalline (Figure R11a,b). During electrode processing, however, TABQ dissolved in NMP and re-precipitated as a thin layer covering carbon particles upon solvent evaporation. It results in short coherent lengths in the TABQ structure and leads to the broad XRD peaks (Figure R11c,d). Nevertheless, the effective interaction between TABQ and carbon ensures good electrical conductivity in electrodes and thus high electrochemical activities. We have added the discussion to the revised manuscript (Page 8 in the revised manuscript and Supplementary Figure 3 in the revised supplementary information).

Figure R11 | The structure of TABQ. a XRD pattern and b SEM image of the as-prepared TABQ. c. XRD pattern and d SEM image of the TABQ electrode.

3. Also, in line 143, it is mentioned that XRD after 1000 cycles verifies the excellent structural stability. But according to my comment above, how this can be confirmed. Because if I compare the XRD data of the pristine TABQ (Fig. S1) and the one after 1000 cycles, besides the similar low-

crystallinity, they are also quite different. In pristine, there are two peaks at 20° and 28°, while after cycling, the peaks at 9° and 27° become pronounced. I would suggest authors to double-check again the XRD data, as for now this cannot support the claim of excellent structural stability.

Response:

We thank the reviewer for the suggestion. We agree with the reviewer that the XRD patterns were not the same between the pristine electrode and after 1000 cycles. The comparison in the original manuscript was made between the electrode after 1st charge and 1000 cycles, and our comment was not very accurate. Comparing the XRD patterns of the pristine electrode and after the 1st charge, the peak at 26.5 degrees increases and the one at 28.3 degrees decreases (Figure R12). It could be attributed to the slight modification of crystal symmetry or π - π stacking manner in the structure of TABQ with the disruption and reconstruction of hydrogen bonding network during the redox processes of carbonyl groups [Chem. Sci. 2019, 10, 418-426; Nucleic Acids Res. 2016, 44, W367-W374]. After 1000 cycles, the peak at 26.5 degrees persists and becomes broader. The small peak below 10° corresponds to $Zn_4SO_4(OH)_6 \cdot 4H_2O$, which is attributed to the small amount of $Zn_4SO_4(OH)_6 \cdot 4H_2O$ remained over long-term cycling. However, due to the broadening of XRD peaks, it is difficult to extract much structural information. We therefore remove the XRD discussion and leave the FT-IR after 1000 cycles to demonstrate the compositional stability of TABQ over cycling.

Figure R12 | XRD comparison of the TABQ electrode at the pristine, 1st charged and 1000th charged states.

4. TABQ is mentioned at the same time to be soluble in water, while also mentioning it's insolubility in the electrolyte (1 M $ZnSO_4$ aqueous electrolyte) in Line 152. Salt concentration can impact the solubility of organics, but not sure it can render completely insoluble. Quantified data would be better to have so if possible, please provide proofs of solubility in the electrolyte, by immersing the compound directly in the electrolyte, eventually by measuring and quantifying this as in DOI: 10.1039/C8SC02995D.

Response:

We thank the reviewer for the suggestion. We carried out UV-vis analysis to quantify the solubility of TABQ in 1 M ZnSO₄ (Figure R13). Standard solutions were prepared with TABQ concentrations of 10, 15, 20 and 45 μmol L⁻¹ in 1 M ZnSO₄. Saturated solutions were obtained by mixing an excess of TABQ in 1 M ZnSO₄ and filtered. The liquid was diluted with 1 M ZnSO₄ by a factor of 100. UV-vis absorbance was measured in the range of 190-600 nm. Calibration curve was obtained by the linear fit of the absorbance at 365 nm with respect to TABQ concentrations of standard solutions using the Beer-Lambert law: $A=\epsilon lc$ (A : absorbance; ϵ : molar extinction coefficient; l : length of the cell; c : TABQ concentration). The concentration of the 100 times diluted saturated solution was calculated from the calibration curve, and the saturated concentration was calculated to be 1.7 mmol L⁻¹ in 1 M ZnSO₄. The low solubility ensures the excellent cycling stability of TABQ. We have added the discussion in the revised manuscript (Page 7-8 and Figure 1f in the revised manuscript).

Figure R13 | UV-vis analysis of TABQ. **a** The UV-vis absorption spectrum of TABQ in 1 M ZnSO₄. **b** UV-vis calibration curve obtained by the linear fit of the absorbance at 365 nm with respect to TABQ concentrations of standard solutions using the Beer-Lambert law $A=\epsilon lc$, together with the calculation of the 100 times diluted saturated solution.

5. For electrochemical measurements, the authors used 40% of KB conductive agent which is a capacitance carbon, capable of provide large amount of capacity depending on the cycling rate. Since it is claimed that TABQ is not soluble in the electrolyte, the authors, if possible, should provide a charge-discharge data using other conductive carbon like Super P or similar? Or a pure KB electrode charge-discharge data can also make this work more rigorous.

Response:

We thank the reviewer for the suggestion. The discharge/charge behaviors of TABQ/KB, TABQ/super P, and KB only in 1 M ZnSO₄ electrolyte are compared (Figure R14). The KB conductive agent delivers less than 25 mAh g⁻¹ capacity without any redox peak in differential capacity curve. It suggests the electrochemical activity of TABQ/KB is mostly from TABQ. With super P as the conductive agent, TABQ delivers around 50 mAh g⁻¹ lower capacity and slightly larger overpotential. It is attributed to the smaller surface area and worse electrical conductivity of super P in comparison to KB. KB was therefore used as the conductive agents for TABQ. We have added the discussion to the revised manuscript (Page 9 in the revised manuscript and Supplementary Figure 4 in the revised supplementary information).

Figure R14 | Electrochemical performance of TABQ/KB, TABQ/super P, and KB electrodes in 1 M ZnSO₄ electrolyte. a Charge/discharge curves and **b** differential capacity curves.

6. Line 198, the formation of Zn₄SO₄(OH)₆·4H₂O is due to the local pH increase. Why there is local pH increase, and why local pH increase will give rise to the formation of Zn₄SO₄(OH)₆·4H₂O. Could the authors give more detailed explanation/discussion in the paper, for example a citation of a previous work, so even a reader from different field can understand and follow easily this work?

Response:

We thank the reviewer for the suggestion. The local pH increase is due to proton insertion into the cathode which leaves OH⁻ behind. Zn₄SO₄(OH)₆·4H₂O precipitation takes place in solutions with pH above 5.5 and dissolves with lower pH [ChemSusChem 2016, 9, 2948-2956; Electrochim. Acta 2012, 85, 438-443]. Therefore, the formation of Zn₄SO₄(OH)₆·4H₂O is widely used as an indicator for proton insertion into cathode materials from ZnSO₄ electrolytes [Nat. Energy 2016, 1, 16039; Nat. Commun. 2018, 9, 1656; Nat. Commun. 2018, 9, 2906]. We have added the discussion and references to the revised manuscript (Page 13 in the revised manuscript).

7. The pH dependent redox potential of TABQ is also slightly ambiguous. In the electrolyte of 1M ZnSO₄ (pH=4), the tetraamine is not able to be protonated to NH₃⁺, while in the 0.5M H₂SO₄ electrolyte, the tetraamine is fully protonated to NH₃⁺. In the case of the latter, NH₃⁺ serves as strong electron withdrawing group, decreasing the electron density of the redox center, and thus should increase the redox potential. So what plausible explanation can be provided for their data by the authors?

Response:

We thank the reviewer for the suggestion. The state of amino groups on TABQ in 1 M ZnSO₄ was studied by UV-vis analysis. In a non-protonated TABQ molecule, the lone pair electrons on the amino groups form extended conjugation with the π electrons on the quinone ring [J. Mol. Struct. 1984, 114, 249-252; J. Am. Chem. Soc. 2002, 124, 2518–2527]. This extended conjugation is disturbed upon the protonation of amino groups. Therefore, the B-band of quinone ring in protonated TABQ possesses similar energy with benzoquinone [Energy Environ. Sci. 2017, 10, 2372] and is blue-shifted in comparison to non-protonated TABQ due to the absence of extended conjugation. Figure R15 shows the UV-vis spectra of TABQ in water and 1 M ZnSO₄. A clear blueshift of B-band from 268 nm in water to 201 nm in 1 M ZnSO₄ is noted, confirming its protonation in the latter. As a result, the amino groups on TABQ are protonated in 1 M ZnSO₄ as well as other more acidic electrolytes, and the redox potential is majorly affected by the proton concentrations based on the Nernst shift. We have added the discussion in the revised manuscript (Page 15-16 and Figure 5a in the revised manuscript).

Figure R15 | UV-vis absorption spectra of TABQ in 1 M ZnSO₄ and water.

REVIEWER COMMENTS

Reviewer #2 (Remarks to the Author):

My questions have been partially addressed in the revision. As detailed below, more explanation is necessary to address them fully. Therefore I recommend rejection based on our current status.

For my previous question 1, authors argued formation of $\text{Zn}_4\text{SO}_4(\text{OH})_6 \cdot 4\text{H}_2\text{O}$ as a key indicator for proton storage of TABQ. While this indirect evidence is helpful to certain degree, quantitative characterization of the stoichiometry of discharge product, i.e. H_xTABQ , is critical to support the proposed reaction equation (1) that x equals to 2, because existence of $\text{Zn}_4\text{SO}_4(\text{OH})_6 \cdot 4\text{H}_2\text{O}$ could also potentially lead to mixed Zn and proton storage. EDS mapping in Fig. R5 indeed points to partial Zn storage for discharged samples.

For my previous question 2, the claim that 'TCBQ, PTO and C4Q do not contain protonation sites' is simply wrong and misleading. It is well known that carbonyl groups on quinones are active protonation sites (DOI: 10.1021/acseenergylett.9b00739 as one example). Therefore, the correlation between the substituted functional groups (-OH, -NH₂) and the conflicting storage modes (Zn²⁺ vs. H⁺ storage) for quinone molecules remains poorly understood in the current manuscript.

Reviewer #3 (Remarks to the Author):

The authors thoroughly addressed my comments. They have provided further experimental evidence to show the potential of the TABQ material for H/Zn storage, which largely addressed my original concerns. They also corrected all the vague expression of electrochemical results. The revised manuscript further elaborated the novelty and impact of the work. I have no further comments and recommend to publish this work.

Dr. Xiaoqi Sun.

Tel: 86-24-83689510

Fax: 86-24-83684323

Email: sunxiaoqi@mail.neu.edu.cn

Response to Reviewers' Comments (Manuscript number NCOMMS-21-00810A)

We thank the reviewers very much for their careful review of our manuscript. We have addressed or clarified the issues raised in the reviewers' reports. Our responses are as follows.

Response to reviewer 2

Comments:

Reviewer #2 (Remarks to the Author):

My questions have been partially addressed in the revision. As detailed below, more explanation is necessary to address them fully. Therefore I recommend rejection based on our current status.

Response:

We thank the reviewer for the careful review of our manuscript and the insightful comments. Our point-to-point responses are listed below.

1. For my previous question 1, authors argued formation of $Zn_4SO_4(OH)_6 \cdot 4H_2O$ as a key indicator for proton storage of TABQ. While this indirect evidence is helpful to certain degree, quantitative characterization of the stoichiometry of discharge product, i.e. H_xTABQ , is critical to support the proposed reaction equation (1) that x equals to 2, because existence of $Zn_4SO_4(OH)_6 \cdot 4H_2O$ could also potentially lead to mixed Zn and proton storage. EDS mapping in Fig. R5 indeed points to partial Zn storage for discharged samples.

Response:

We thank the reviewer for the comment. We agree with the reviewer that quantitative analysis is necessary to identify the amount of proton insertion vs. Zn^{2+} insertion into TABQ. We carried out

inductively coupled plasma optical emission spectrometry (ICP-OES) and ion chromatography (IC) on the discharged cathode to quantify the weight percentages of zinc and sulfate, respectively. They result in Zn wt% and SO_4^{2-} wt% of 17.34% and 5.76% in the discharged cathode, respectively. The SO_4^{2-} exists in $\text{Zn}_4\text{SO}_4(\text{OH})_6 \cdot 4\text{H}_2\text{O}$, and Zn exists in $\text{Zn}_4\text{SO}_4(\text{OH})_6 \cdot 4\text{H}_2\text{O}$ as well as Zn^{2+} inserted TABQ. The discharged cathode contains the mixture of: (1) proton inserted TABQ (2H-TABQ), (2) Zn^{2+} inserted TABQ (Zn-TABQ), (3) $\text{Zn}_4\text{SO}_4(\text{OH})_6 \cdot 4\text{H}_2\text{O}$ and (4) KB and PVDF. Their weight percentages obey the following equations:

$$2\text{H-TABQ wt\%} + \text{Zn-TABQ wt\%} + \text{Zn}_4\text{SO}_4(\text{OH})_6 \cdot 4\text{H}_2\text{O wt\%} + [\text{KB} + \text{PVDF}] \text{ wt\%} = 100\% \quad (1)$$

$$\text{SO}_4^{2-} \text{ wt\%} = \text{Zn}_4\text{SO}_4(\text{OH})_6 \cdot 4\text{H}_2\text{O wt\%} \times \omega[\text{SO}_4^{2-} \text{ in } \text{Zn}_4\text{SO}_4(\text{OH})_6 \cdot 4\text{H}_2\text{O}] = 5.76\% \quad (2)$$

$$\text{Zn wt\%} = \text{Zn-TABQ wt\%} \times \omega[\text{Zn in Zn-TABQ}] + \text{Zn}_4\text{SO}_4(\text{OH})_6 \cdot 4\text{H}_2\text{O wt\%} \times \omega[\text{Zn in } \text{Zn}_4\text{SO}_4(\text{OH})_6 \cdot 4\text{H}_2\text{O}] = 17.34\% \quad (3)$$

$$\text{TABQ wt\%} = 2\text{H-TABQ wt\%} \times M_w(\text{TABQ})/M_w(2\text{H-TABQ}) + \text{Zn-TABQ wt\%} \times M_w(\text{TABQ})/M_w(\text{Zn-TABQ}) = [\text{KB} + \text{PVDF}] \text{ wt\%} \quad (4)$$

The meaning of symbols in the above equations are as below:

2H-TABQ wt%: weight percentage of proton inserted TABQ in the discharged cathode;

Zn-TABQ wt%: weight percentage of Zn^{2+} inserted TABQ in the discharged cathode;

$\text{Zn}_4\text{SO}_4(\text{OH})_6 \cdot 4\text{H}_2\text{O}$ wt%: weight percentage of $\text{Zn}_4\text{SO}_4(\text{OH})_6 \cdot 4\text{H}_2\text{O}$ in the discharged cathode;

[KB + PVDF] wt%: weight percentage of KB and PVDF in the discharged cathode;

$\omega(\text{SO}_4^{2-} \text{ in } \text{Zn}_4\text{SO}_4(\text{OH})_6 \cdot 4\text{H}_2\text{O})$: mass fraction of SO_4^{2-} in $\text{Zn}_4\text{SO}_4(\text{OH})_6 \cdot 4\text{H}_2\text{O}$, which is 18.07%;

$\omega(\text{Zn in } \text{Zn}_4\text{SO}_4(\text{OH})_6 \cdot 4\text{H}_2\text{O})$: mass fraction of Zn in $\text{Zn}_4\text{SO}_4(\text{OH})_6 \cdot 4\text{H}_2\text{O}$, which is 49.19%;

$\omega[\text{Zn in Zn-TABQ}]$: mass fraction of Zn in Zn^{2+} inserted TABQ, which is 28.00%;

$M_w(\text{TABQ})$: molecular weight of TABQ, which is 168.15 g mol⁻¹;

$M_w(2\text{H-TABQ})$: molecular weight of proton inserted TABQ, which is 170.17 g mol⁻¹;

$M_w(\text{Zn-TABQ})$: molecular weight of Zn^{2+} inserted TABQ, which is 233.54 g mol⁻¹.

The above equations result in the 2H-TABQ and Zn-TABQ weight percentages of 29.14% and 5.93%, respectively. Therefore, the mole of inserted proton is 13.5 times of Zn^{2+} . It confirms the domination of proton storage in TABQ during the redox processes. The above discussion has been added to the revised manuscript (page 14-15 in the revised manuscript and page 11-12 in the revised supplementary information).

2. For my previous question 2, the claim that 'TCBQ, PTO and C4Q do not contain protonation sites' is simply wrong and misleading. It is well known that carbonyl groups on quinones are active protonation sites (DOI: 10.1021/acsenergylett.9b00739 as one example). Therefore, the correlation between the substituted functional groups (-OH, -NH₂) and the conflicting storage modes (Zn²⁺ vs. H⁺ storage) for quinone molecules remains poorly understood in the current manuscript.

Response:

We thank the reviewer for the comment. We agree with the reviewer that our previous expression about protonation sites was not precise. Nevertheless, the carbonyl groups on quinones possess negative pK_a values below -6, which means the protonation is negligible in the ZnSO₄ electrolyte with pH around 4 [B. Chem. Soc. Jpn. 1955, 28, 483-489; Electrochim. Acta. 1968, 13, 721-749; J. Electroanal. Chem. 1984, 164, 213-227]. We also checked the paper proposed by the reviewer [DOI: 10.1021/acsenergylett.9b00739], and the protonation discussed there was based on the reduced carbonyl groups. We have revised our previous claim of 'TCBQ, PTO and C4Q do not contain protonation sites' into 'TCBQ, PTO and C4Q would not protonate in the weakly acidic zinc electrolytes' to be more precise (page 17 in the revised manuscript).

Response to reviewer 3

Comments:

Reviewer #3 (Remarks to the Author):

The authors thoroughly addressed my comments. They have provided further experimental evidence to show the potential of the TABQ material for H/Zn storage, which largely addressed my original concerns. They also corrected all the vague expression of electrochemical results. The revised manuscript further elaborated the novelty and impact of the work. I have no further comments and recommend to publish this work.

Response:

We thank the reviewer for the careful review of our manuscript and the recommendation to publish our work.